# The Impact of MicroRNAs during Inflammatory Bowel Disease: Effects on the Mucus Layer and Intercellular Junctions for Gut Permeability

**DOI:** 10.3390/cells10123358

**Published:** 2021-11-30

**Authors:** Sarah Stiegeler, Kevin Mercurio, Miruna Alexandra Iancu, Sinéad C. Corr

**Affiliations:** 1Department of Microbiology, Moyne Institute of Preventative Medicine, School of Genetics and Microbiology, Trinity College Dublin, Dublin, Ireland; stiegels@tcd.ie (S.S.); MERCURIK@tcd.ie (K.M.); IANCUM@tcd.ie (M.A.I.); 2APC Microbiome Ireland, University College Cork, Cork, Ireland

**Keywords:** microRNAs, inflammatory bowel disease, gut epithelial barrier, mucus layer, intercellular junctions

## Abstract

Research on inflammatory bowel disease (IBD) has produced mounting evidence for the modulation of microRNAs (miRNAs) during pathogenesis. MiRNAs are small, non-coding RNAs that interfere with the translation of mRNAs. Their high stability in free circulation at various regions of the body allows researchers to utilise miRNAs as biomarkers and as a focus for potential treatments of IBD. Yet, their distinct regulatory roles at the gut epithelial barrier remain elusive due to the fact that there are several external and cellular factors contributing to gut permeability. This review focuses on how miRNAs may compromise two components of the gut epithelium that together form the initial physical barrier: the mucus layer and the intercellular epithelial junctions. Here, we summarise the impact of miRNAs on goblet cell secretion and mucin structure, along with the proper function of various junctional proteins involved in paracellular transport, cell adhesion and communication. Knowledge of how this elaborate network of cells at the gut epithelial barrier becomes compromised as a result of dysregulated miRNA expression, thereby contributing to the development of IBD, will support the generation of miRNA-associated biomarker panels and therapeutic strategies that detect and ameliorate gut permeability.

## 1. Introduction

Since their discovery in 1993 [1,2], microRNAs (miRNAs) were shown to play critical roles in various biological processes. MiRNAs are small, non-coding RNAs of 18–24 nucleotides (nt) in length known to interfere with RNAs. They are most commonly described in the literature for their interaction with mRNAs whereby they fine-tune protein synthesis at the translational level. The details of miRNA biogenesis have been extensively reviewed previously [3,4,5,6] and will only be summarised here. Briefly, miRNAs are encoded within intergenic, intronic and exonic regions of the human genome [7], with the majority of miRNAs found in the intronic regions of both protein-coding and non-coding genes [8]. Their biogenesis starts with a primary (pri)-miRNA molecule of nuclear, hairpin-structure. The pri-miRNA matures through sequential steps of enzymatic cleavage, first in the nucleus by Drosha/DGCR8 and then in the cytoplasm by Dicer, leaving a processed miRNA duplex in the cytoplasm. Finally, the guidance strand of the miRNA duplex is loaded into an Argonaute (AGO) protein, forming the RNA-induced silencing complex (RISC) [3,4]. While both complimentary strands of the miRNA duplex can be loaded into the AGO protein and exhibit bioactivity, typically only one of the two miRNAs will show predominant activity. RISC-targeted RNA molecules are recognised by sequence-specific binding and typically lead to the silencing of mRNA. In mammals, the miRNA seed region (~2–7 nt) is the dominant motif for target recognition and miRNA–mRNA binding. Yet, additional binding sites may improve binding to its target mRNA [6,9]. Friedman et al. reported that over 60% of human protein-coding genes have a predicted, well-conserved 3′ binding site in their untranslated region (UTR) for miRNAs [10]. In addition, miRNA binding motifs are found in protein-coding sequences and at the 5′-UTR, but miRNA-mediated mRNA repression seems to be most efficient when binding to the 3′-UTR [11,12].

The actions of miRNAs are highly cell- or tissue-specific. Furthermore, several miRNAs can target one mRNA simultaneously to enhance their action, or vice versa, one miRNA can target several different mRNAs. While there are some cases of miRNA-mediated activation of protein synthesis [3,13], the miRNA–mRNA interaction typically leads to a repression of translation for the target. Whether the binding of an miRNA to its target mRNA leads to the inhibition of the translational process or to mRNA degradation is determined by the specific binding capacity. Hence, the combination of a perfect match between the seed region base-pairing to the central region of the miRNA leads to mRNA degradation. In contrast, imperfect binding of the seed region is typically associated with translational inhibition [5]. Consequently, dysregulated miRNA expression profiles are often correlated or may even be the cause of a plethora of human diseases. Several studies showed modulated miRNA expression profiles in cancer [14], cardiovascular diseases [15] and chronic inflammatory disorders such as inflammatory bowel disease (IBD) [16,17]. Thus, miRNAs have a large impact on the regulation of inflammation from a disease context. Indeed, miRNAs are essential for the proper functioning of cellular pathways involved in gastrointestinal (GI) health, such as cell differentiation, proliferation, apoptosis and more broadly the innate and adaptive immune response to microbiota [18]. In this review, we have summarised the key research on the immunological roles of miRNAs relevant to IBD, their potential uses as diagnostic biomarkers and treatments, and focus on their role in gut permeability.

## 2. MicroRNAs and Disease

### 2.1. Inflammatory Bowel Disease

IBD is a debilitating autoimmune disease characterised by chronic inflammation along the GI tract. Patients diagnosed with IBD are symptomatic for recurrent intestinal inflammation, diarrhoea, abdominal pain, rectal bleeding, weight loss and anaemia. Due to its complexity, a number of factors are attributed to IBD aetiology, including patients’ genetics and makeup of microbiota, food and pharmaceutical consumption, and even limiting antigen exposure due to excessive sanitation [19,20]. All these aspects further contribute to changes in miRNA expression.

IBD is caused by the overactivation of the mucosal immune system driven mainly by increased exposure to the gut microbiota as a result of compromised gut permeability. Importantly, host genetics are a major factor attributed to the manifestation of disease and linked to multiple regions of the genome [20]. For example, it is more likely for IBD patients to possess variants of the *NOD2*/*CARD15* gene on chromosome 16 than healthy individuals, with this gene encoding a specific pattern-recognition receptor (PRR) for bacterial lipopolysaccharide that regulates macrophage activation of nuclear factor-κB (NF-κB) [21]. Additionally, researchers consistently observe that the IBD3 gene on chromosome 6, which encodes the major histocompatibility complex (MHC), has a genetic linkage to IBD [21]. Other environmental factors that increase IBD incidence include the use of non-steroidal anti-inflammatory drugs, antibiotics and smoking, which are all known to alter the gut microbiota [22].

Clinically, IBD is segregated into two main types known as Crohn’s disease (CD) and ulcerative colitis (UC). CD can impact any part of the GI tract with patchy regions of inflammation, whereas UC has inflammation typically localised to the colon or rectum [23]. Incidence of CD or UC can also lead to an increased risk of other diseases such as colorectal cancer [24]. The activation of central immune cell populations leads to the recruitment of non-specific mediators of the inflammatory response, such as the formation of metabolites such as prostaglandins and leukotrienes, along with damaging compounds such as reactive oxygen species (ROS) [20]. All these factors can compromise the gut epithelial barrier where the majority of host and gut microbiota interactions occur. In such a disease state, luminal antigens gain access to the lamina propria, triggering a response by innate and adaptive immune cells via various PRRs, causing professional (i.e., dendritic cells) and non-professional (i.e., intestinal epithelial cells, IECs) antigen presenting cells (APCs) to further activate central effector immune cells as well as other pro-inflammatory mediators. This cascade thereby perpetuates a positive feedback loop of leukocyte recruitment and increasing tissue damage in both types of IBD [19].

Research on IBD has highlighted changes in gut immunity and the overall functioning of cells that participate in the characteristic overactive immune response. Immune cells can be a part of innate immunity, which serves as the initial rapid defence upon the recognition of foreign pathogens, or adaptive immunity, which leads to slow but long-lasting defensive measures [25]. An effective inflammatory response to any pathogenic invasion is conducted mainly by innate immune cells, which include neutrophils, dendritic cells, monocytes, macrophages and natural killer cells. For an immunological response to specific pathogens, adaptive immune cells are more critical and include effector T-cells, regulatory T-cells and B-cells [25]. The function within, and communication between, the two arms of immunity contribute to chronic inflammation associated with IBD, leading to tissue damage through ROS production, fibrosis and continuous feedback loops of pro-inflammatory cytokine signalling [19].

There are notable immunological differences between CD and UC. Previously accepted notions were that CD typically has a major CD4+ lymphocyte population with a type-1 helper T-cell (Th1) phenotype, while UC has a type-2 helper T-cell (Th2) phenotype [26]. Due to these differences in lymphocyte populations, CD is driven by interferon-γ (IFN-γ) and interleukin-12 (IL-12) expression, while UC is driven by transforming growth factor (TGF)-β, IL-4, IL-5 and IL-13 expression. More recently, this paradigm has been expanded to incorporate the IL-23/Th17 axis to further distinguish the two types [26]. The interactions between immune cells and cells that constitute the gut epithelial barrier are paramount in understanding permeability within CD and UC disease states.

### 2.2. Biomarkers and Treatments

The archetypical method of diagnosis for IBD is endoscopy in conjunction with biopsies [27]. Despite the established belief that CD is mainly attributed to the overactivation of the inflammatory response throughout the GI tract while UC is mainly confined to the rectum and colon, there remain issues in the proper diagnosis of the two conditions. Serological biomarkers include the presence of perinuclear staining anti-neutrophil cytoplasmic antibodies in 70% of UC patients, or the presence of anti-*Saccharomyces cerevisiae* antibodies in 50% of CD patients [28]. Other biomarkers include C-reactive protein in serum and granulocyte proteins lactoferrin and calprotectin in faeces [29]. Recently, a panel of 51 protein biomarkers in serum was determined to be predictive of CD within up to five years before diagnosis that included associated changes to complement cascade, lysosomes, innate immune response and glycosaminoglycan metabolism [30]. Further development on IBD biomarkers is still required for differentiation between CD and UC, along with indication of their severity and active state.

Due to their high stability in bodily fluids, miRNAs have been studied as potential biomarkers of disease [31]. Research has expanded novel diagnostic tools to determine differences in the presence of miRNAs within various samples, with miRNAs being observed in serum, urine, tears, saliva and breast milk [32]. Total serum RNA from active and inactive CD and UC patients demonstrated that miR-595 and miR-1246 were significantly upregulated in IBD and could serve as non-specific biomarkers [33]. In plasma samples, only significant downregulation of miR-16 was validated for diagnosing CD [34]. Whole blood samples obtained from IBD patients showed that CD4+ T-cell expression of miR-1307-3p, miR3615 and miR-4792 predicted disease progression of IBD [35]. In testing miRNA profiles of saliva samples taken from IBD patients, there was a significantly altered expression of miR-101 in CD patients and miR-21, miR-31, miR-142-3p and miR-142-5p in UC [36]. Interestingly, freshly frozen colonic mucosa tissues from IBD patients displayed high levels of miR-31, miR-146a, miR-206 and miR-424, with miR-31 also highly expressed in formalin-fixed, paraffin-embedded tissues [37]. A recent study investigated the mucosal and serum expression of miRNAs in the colon of a canine IBD model. In canine IBD, miR-16, miR-21, miR-122 and miR-147 were elevated in serum and the colonic mucosa, while miR-146a, miR-192 and miR-223 were upregulated in the serum only compared to their controls [38]. In human blood or biopsy samples of IBD patients, among others, an elevation of miR-16, miR-21, miR-106a, miR-122, miR-151-5p, miR-155, miR-199a-5p, miR-320 and miR-362-3p was observed [16,39,40,41]. The variety of samples in which miRNAs exist provide several options for studying biomarkers in IBD.

There remains an emphasis on biomarker studies that show distinguishable aspects between CD and UC for novel diagnostic tools. Some research has looked at developing miRNA panels for CD and UC diagnosis with great accuracy. One study identified an 11-miRNA panel for CD using serum samples [42], while another used platelet-derived miRNAs to determine a 31-miRNA panel for UC [43]. A further study used a six-miRNA panel to distinguish between CD and UC from colon biopsies [36]. Using peripheral blood, an eight-miRNA panel was found to distinguish between active UC and CD [44]. In addition, differentiation between types of IBD and intestinal colitis is also crucial in furthering diagnostic methods. Differential expression of miR-24 allowed researchers to distinguish between UC and L2 CD within rectal biopsies [45]. These advances are necessary for identifying specific treatments tailored uniquely to the patient.

Determining IBD activity is also crucial in prescribing treatments and predicting patient health impacts. One study found that miR-150, miR-196b, miR-199a-3p, miR-199b-5p, miR-223 and miR-320a displayed significant differential expression in non-inflamed UC compared to non-inflamed CD colonic tissues [46]. Another study demonstrated that miR-20b, miR-26b, miR-98, miR-99a and miR-203 were significantly upregulated in colonic mucosal pinch biopsies obtained from patients with active UC compared to quiescent UC [47]. Significant downregulation of miR-192, miR-375 and miR-422b and a significant upregulation in miR-16, miR-21, miR-23a, miR-24, miR-29a, miR-126, miR-195 and let-7f have been observed in sigmoid colon pinch biopsies [48]. Levels of miR-192 were significantly upregulated and miR-16 significantly downregulated in active UC [48,49]. A downregulation in miR-4284 in colonic tissue samples from active UC patients was also observed in a separate study [50]. Moreover, levels of miR-142-5p, miR-595 and miR-1246 in serum samples could differentiate active and non-active CD with high accuracy [33]. Interestingly, miR-31-5p and miR-203 were identified as inflammation-independent diagnostic markers for CD in colonic tissue samples, while miR-215 predicted a specific penetrating/fistulising CD phenotype in the ileum [51]. An overview of altered expression patterns of miRNAs in the context of IBD can be found in Table 1.

Treatments for IBD include pharmaceuticals, antibody therapies and full surgical procedures. One frequently prescribed treatment is 5-aminosalicylate, which was suggested to block the production of prostaglandins and leukotrienes, inhibit bacterial peptide-induced neutrophil chemotaxis, scavenge circulating ROS and further inhibit the activation of NF-κB [20]. Others include the use of corticosteroids, though it is important to note that continuous monitoring of prevalent side effects such as corticosteroid-induced osteoporosis, hypertension or diabetes must be undertaken [20]. Immunosuppressants such as azathioprine are also prescribed; however, this often puts patients at risk of opportunistic infections and toxic side effects such as neutropenia, pancreatitis and drug-induced hepatotoxicity [20]. Distinguishing between types of IBD is crucial as immunoregulatory treatments could sometimes be beneficial for one and not the other. This is indeed the case for cyclosporin in treating UC, or anti-tumour necrosis factor (TNF) therapy via infliximab combined with thiopurines for CD [20,27]. Even appendectomies have been associated with an increased risk of developing strictures in CD patients while having improved effects in UC patients [19].

MiRNAs have been investigated as new therapeutic targets and indicators of drug treatment suitability. For example, utilising miR-200c-3p mimics may reduce levels of inflammation caused by IL-8 or NF-κB response to TLR4 activation in IBD [52]. Overexpression of miR-122 was suggested to downregulate NOD2 in IECs, inhibiting apoptosis and destruction of the intestinal barrier [53]. Inhibiting miR-155 expression may be a therapeutic avenue due to its targeting of SOCS1 in UC or the inhibition of the hypoxia-inducible factor (HIF)-1α/trefoil factor (TFF)-3 axis observed in animal models [54,55]. Moreover, it was suggested that using miR-195 as a biomarker may help track therapeutic steroid resistance in IBD patients [56]. Other miRNAs such as let-7d, let-7e, miR-28-5p, miR-221 and miR-224 were found to be significantly increased in patients after six weeks of infliximab treatment, with let-7d and let-7e also showing upregulation in patients undergoing remission [57]. Further details on other miRNAs studied for their regulation in response to specific therapies and as treatment biomarkers are described elsewhere [58].

### 2.3. Gut Immunity

Research on IBD and gut immunity uses multiple in vivo models. For example, the dextran sulfate sodium (DSS) mouse model pertains more closely to UC with colitis manifesting due to destruction of the intestinal barrier, while the T-cell adoptive transfer mouse model more closely resembles CD with a focus on early immunological factors [31]. Interestingly, drastic impacts occur upon the deletion of Dicer1 within the intestinal epithelium of mice, one of the major enzymes in miRNA biogenesis, including the spontaneous development of colitis and a dramatic increase in epithelial cell apoptosis [59], highlighting the relevance of miRNA-mediated regulation of the GI barrier. Other chemically induced or knockout (KO) mouse models, along with alternative model organisms used in the study of IBD pathogenicity, are well-described elsewhere [60,61].

A difficulty in studying miRNA relevance in IBD and innate immunity is the species- and cell-specific features. For instance, Toll-like receptors (TLRs) are key for IECs to identify pathogen-associated molecular pathways (PAMPs). TLR4-activated NF-κB induction of microRNA-9 (miR-9) occurs primarily in human monocytes and neutrophils, while miR-210 plays a regulatory role in the NF-κB feedback pathway typically in murine macrophages [62,63]. Additionally, nucleotide-binding and oligomerization domain (NOD)-like receptors (NLRs) are important in the surveillance of the intracellular environment for signs of possible infection. Modulations by miR-29 in general dendritic cells, miR-122 in HT-29 cells, miR-146a in muramyl dipeptide (MDP)-activated macrophages and miR-192/miR-495/miR-512/miR-671 in HCT116 cells were shown to play critical roles in the regulation of NLRs [53,64,65,66]. It is essential that researchers understand the limitations to their study models and explore all alternatives to study miRNA relevance in IBD.

One of the most widely studied miRNAs regarding health and disease is miR-21. Among the most abundantly expressed miRNAs in various mammalian cell types, miR-21 is considered an oncomiR within the intronic region of the protein coding gene *TMEM49* [67]. The regulation of miR-21 is still not fully understood, as there are multiple layers towards maturity that can be regulated, including several transcription factors that bind to its promoter region or bind to the pri-miR-21 form [68]. Elevated levels of miR-21 are suggested to be pathological in IBD [69,70]. Epigenome-wide whole blood DNA methylation profiles of paediatric CD treatment-naïve patients showed that hypomethylation of the miR-21 locus correlated with increased expression in leukocytes and inflamed intestinal tissue [71]. Importantly, several studies showed that the ablation of miR-21 in mice led to protection against DSS-induced colitis [72,73]. For UC patients in remission, miR-21 was found to be downregulated while known target programmed cell death protein (PDCD)-4 was upregulated in CD3+ T-cells compared to active disease and healthy controls [74]. Further work is required to understand the full scope of miR-21 influence within the inflamed gut.

Other miRNA KO models demonstrated amelioration during DSS-induced colitis. Like miR-21, the deletion of miR-155 in mice protected against DSS-induced colitis [75]. Additionally, the deletion of miR-301a also protects mice against DSS-induced colitis by rescuing BTG anti-proliferation factor 1 (BTG1) expression and is associated with lowering levels of pro-inflammatory markers such as IL-1β, IL-6, IL-8 and tumour necrosis factor (TNF) [76]. Genetic studies on the consequences of combined KO models could elaborate more on the negative roles these miRNAs have in IBD.

Another hallmark factor in those afflicted by IBD is the presence of oxygen, either as increased levels of ROS through continuous activation of macrophages or the sensing of oxygen in the gut environment. Several miRNAs were shown to be involved in regulating nitric oxide synthase-2 (NOS2) in IBD tissues. Induction of the nitric oxide pathway by miR-21, miR-126, miR-146a, miR-221 and miR-223 led to senescence among adjacent epithelial cells via the upregulation of HP1γ [77]. Regarding the sensing of environmental oxygen in the gut, HIF was demonstrated to be a key regulator of barrier integrity and induced expression of miR-320a to improve barrier function in T84 cells [78]. Developing methods for measuring levels of oxygen as damaging free radicals and in its gaseous state within the inflamed gut may help researchers track the progression of the disease.

Numerous miRNAs demonstrated relevance to IBD and adaptive immunity. T-cells have significant roles in the genesis and development of IBD. The deletion of miR-21 exacerbates CD4+ T-cell-mediated models of colitis, while loss of miR-155 tends to decrease Th1/Th17, showing that these are key regulators in regulatory T-cell (Treg) homeostasis [75,79]. Continued work has demonstrated other miRNAs in Treg regulation such as miR-10a, miR-17-92 cluster, miR-146a and miR-212/132 [80,81,82,83]. MiRNAs implicated in Th1 and Th2 differentiation include miR-17-92 cluster, miR-27b, miR-29, miR-128, miR-146a, miR-155 and miR-340 [82,84,85,86]. For Th17 differentiation, miRNAs that show impact when imbalanced are miR-10a, miR-155 and miR-326, as well as miR-301a as an indirect inducer [87,88,89,90]. Overexpression of miR-210 may negatively impact Th17 differentiation due to targeting hypoxia-induced inhibitor HIF1α [91]. Finally, B-cell maturation was shown to be regulated by miR-10a, miR-17-92 cluster and miR-181a [87,92,93]. The coordinated interplay between immunity regulation and IECs is essential in controlling barrier permeability. Further information on relevant research models used in the study of IBD and their conclusions regarding miRNAs’ impact on pathogenesis has been summarised elsewhere [94].

## 3. Permeability of the Gut Epithelial Barrier

Under normal circumstances, permeability of the gut epithelial barrier is warranted since it is at this interface that essential nutrients can be absorbed and taken up into the human body. The gut is also a region where many symbiotic microbes reside, contributing to the breakdown of food, competing with pathogenic invaders and priming our GI-associated immune system for external threats. However, impaired or increased permeability has been associated with IBD pathogenesis. Bischoff et al. define the term as “a disturbed permeability being non-transiently changed compared to the normal permeability leading to a loss of intestinal homeostasis, functional impairments and disease” [95]. There are several external factors that can contribute to impaired permeability of the gut epithelial barrier including dietary regimen, pharmaceuticals, smoking, as well as physical cellular factors such as the immune system, presence of microbiota, the mucus layer and IEC adherence and communication via intercellular junctions [18,95,96]. Additionally, cellular processes such as autophagy and the epithelial–mesenchymal transition have also been implicated in IBD pathogenesis [31,58,97]. Importantly, all these cellular factors can be regulated by miRNAs. Here, we focus on two components comprising the initial physical barrier that governs permeability during IBD, the mucus layer and intercellular junctions of IECs, and discuss the current understanding of how miRNAs regulate their functions.

### 3.1. Protection by the Gut Mucosa

#### 3.1.1. General Characteristics

One of the primary functions of the gut is to digest and absorb nutrients. Most nutrients are absorbed in the small intestine, passing on pellets to the colon that contain nutrients exceeding the absorption capacity of the small intestine along with indigestible fibres. Importantly, complex carbohydrate structures are one of the main exogenous energy sources for the colonic microbiota. The variety of fibres from different sources can support the diversity within the gut microbiota and thereby support gut health [98].

Forming the gut lining, IECs are organised into macrostructures called villi (small intestine only) and crypts (small intestine and colon). Every 4–5 days, the epithelial lining of the gut is completely renewed [99]. Intestinal stem cells are located in the crypts, pushing newly differentiated cells upwards and thereby maintaining the epithelium. The gut epithelium consists of highly specialised cells, such as enterocytes, enteroendocrine cells, Tuft cells, goblet cells and Paneth cells. Enterocytes are involved in cell–cell communication, the absorption of nutrients and the sampling of luminal antigens along the GI tract [100]. Enterocytes also produce membrane-anchored mucins at the apical site, creating a protective cover called the glycocalyx [101]. Paneth cells are mostly found in the small intestine and with a reduced cell number in the proximal colon that give rise to antimicrobial agents. These antimicrobial peptides counteract bacterial growth near IECs [102] and are crucial to limit bacterial exposure in order to maintain the gut barrier. In contrast, intestinal Tuft cells are important for parasitic defence [103]. While enteroendocrine cells secrete hormones, goblet cells are known to secrete gel-forming mucins covering the epithelium in the GI tract [104].

While researchers have a basic understanding of the general aspects governing the gut epithelial barrier, there are still several open questions concerning the underlying molecular mechanisms, such as the dynamic crosstalk between host–microbiota cells and host–host cells, along with overall regulatory circuits. A healthy gut is hallmarked by a functional gut barrier and a diverse gut microbiota. A normal community of microbiota is complex and built from bacteria, viruses, fungi and even archaea [105,106]. In contrast, for IBD patients, the homeostasis of the gut barrier is disrupted and is associated with dysbiosis [107]. The increased permeability of the epithelium observed in patients suffering from CD and UC is one of the main contributors to IBD pathogenicity [107]. The “leaky gut” or increased permeability phenotype in IBD is a direct consequence of a physiological weakening of both the mucus layer and the integrity of the epithelium. A subsequent increased exposure of luminal antigens and microorganisms to the underlying immune system results in inflammation and IBD progression.

In the gut, the first level of the intestinal barrier is the mucus layer, shielding the luminal microbiota from direct contact with the epithelium and the underlying immune system. Here, we highlight how miRNAs might influence the gut barrier at the mucus level and the impact of this on the gut bacterial microbiota.

#### 3.1.2. The Mucus Layer

The mucus layer would not exist without the proper functioning of goblet cells. Goblet cells are the main mucus-producing cells within the GI tract and are essential to build the protective mucus layer. Research has shown that there are distinct differences when comparing the mucus layer between the small intestine and the colon. Mucus of the small intestine forms a single disrupted layer and mainly consists of the mucin MUC2. Upon secretion from goblet cells, MUC2 is attached to the cell membrane, and following enzymatic cleavage by luminal meprin β, it is released from the cell membrane to form the protective mucus cover [108]. This enzymatic release is dependent on a preceding unfolding of MUC2 that allows accessibility to the enzymatic cleavage site [108,109]. It was shown that bacterial-sized beads could penetrate the mucus layer in the small intestine [110], indicating that mucin in the small intestine is loosely organised and does not solely prohibit bacteria from reaching the epithelium. Instead, the mucus layer of the small intestine builds a diffusion barrier for antigens on the luminal side and concentrates antimicrobial agents secreted by Paneth cells at the epithelium. With this, the relative concentrations of antimicrobial peptides such as defensins, REG3γ and lysosomes are relatively high and in close proximity to the epithelium [98]. Furthermore, the release of meprin β was reported to be triggered by microbial sensing, which traps microbes in the mucus to “flush” them away from the epithelium [108].

Even though mucus in both the small and large intestine is based on MUC2, mucus in the colon forms two distinct layers [111]. Johansson et al. reported that the inner mucus layer is attached to the underlying epithelium and forms filter-like structured sheets of MUC2. These structures prohibit direct contact between the bacterial microbiota and the epithelium by size exclusion [111]. In contrast, the outer mucus layer is loose in structure and can be penetrated by bacteria [111]. Thus, this outer layer forms the replicative niche for mucosa-associated microbiota.

Studies have looked at understanding the role of the mucus layer in gut barrier permeability and inflammation. MUC2-deficient (*Muc2^−/−^*) mouse models confirmed the importance of the physical barrier. Due to reduced protection by the mucus layer, the epithelium of *Muc2^−/−^* mice is in close contact with gut microbiota, allowing bacteria to enter the sensitive crypts and trigger inflammation. Thus, *Muc2^−/−^* mice are reported to suffer from severe dysbiosis, develop spontaneous colitis and are prone to colorectal cancer [111,112,113]. In humans, the weakening of the MUC2-dependent inner layer was associated with UC [114]. Patients can suffer from chronic inflammation caused by the commensal microbiota of the mucus layer [114,115].

#### 3.1.3. MicroRNAs, Goblet Cells and Mucus Secretion

How miRNAs are involved in the regulation and secretion of intestinal mucus is largely unknown. There is strong evidence that dysregulated miRNAs have a severe impact on the intestinal mucus barrier. Here, we illustrate the predicted effect of dysregulated miRNAs associated with IBD pathogenicity on mucus components, which contribute to the increased permeability of the gut barrier.

In IBD patients, a lower amount of goblet cells was observed in the upper crypts, with UC patients having even lower levels compared to CD patients [41]. This could be attributed to the high turnover of epithelial cells in the colon, requiring a constant need for goblet cell differentiation and maturation, especially upon inflammation-induced tissue damage. The differentiation of goblet cells is controlled by a Notch-dependent pathway, and the terminal differentiation involves Krüppel-like transcription factor 4 (KLF4), growth factor independence 1 (GFI1) and SAM pointed domain-containing ETS transcription factor (SPDEF) [116,117]. Hath1, a basic helix–loop–helix transcription factor, is needed to counter the differentiation towards absorptive cell development [118]. Increased goblet cell differentiation was observed during inflammation for CD patients but not for UC patients, with levels of HATH1 and KLF4 correlating with mucus production in IBD [118]. According to the predictive database miRWalk [119], all four differentiation markers are predicted to be targeted by IBD-associated miR-16, miR-106, miR-21 (excluding KLF4 and HATH1), miR-122 (excluding KLF4), miR-146, miR-151, miR-155 (excluding KLF4, GFI1 and SPDEF), miR-199 (excluding GFI1) and miR-362 (excluding GFI1 and HATH1). Although these interactions still need to be verified, the pathological levels of these miRNAs might be one explanation for the overall depletion of goblet cells in IBD. Gersemann et al. reported an increased level of goblet cell differentiation factors [118], but in comparison to healthy subjects, the overall goblet cell density remains compromised and might be the result of interference by miRNAs.

The alteration of mucus components in UC patients was investigated by Van der Post et al. Together with MUC2, structural components such as Fc-gamma binding protein (FCGBP), SLC26A3/DRA (downregulated in adenoma) and Zymogen granule protein 16 (ZG16) are found to be reduced in the colonic mucus of UC patients [114]. FCGBP, a core mucus protein, is produced and secreted by goblet cells. Although its function is still not fully understood, it was suggested that FCGBP might act as a mucus cross-linker [120]. Moreover, ZG16 binds and aggregates Gram-positive bacteria and thereby limits their close contact with the epithelium [121]. SLC26A3/DRA, a chloride-bicarbonate transporter, might be involved in a bicarbonate-dependent unfolding of MUC2 upon release [114]. According to the predictive database miRWalk [119], all three mucus components harbour predictive binding sites for miR-16, miR-21, miR-106, miR-146, miR-151, miR-155 (excluding DRA) and miR-362 (excluding SLC26A3/DRA). The elevated levels of the listed miRNAs might impair the mucus layer by a general downregulation of structural mucus components in IBD as well as the exhaustion of sentinel goblet cells, as proposed by Van der Post et al. [114]. SLC26A3/DRA is also negatively affected by the pro-inflammatory cytokine TNF, which was recently reported to play a crucial role in maintaining the gut epithelial barrier [122,123]. Recently, SLC26A3/DRA was downregulated through direct targeting of miR-494 in IECs [124]. Conversely, decreased miR-494 levels have been associated with an increased gut permeability and the severeness of IBD [125], suggesting that the reduced levels of SLC26A3/DRA might not be caused by miR-494 in IBD.

A recent study investigating miRNA expression levels of UC patients gave further insights into a potential miR-21-related loss of mucus integrity [126]. The authors also identified, among others, the upregulation of miR-429 in diseased tissue. Mo et al. further demonstrated that miR-429 directly targets myristoylated alanine-rich protein kinase C substrate (MARCKS) and thereby indirectly downregulates the secretion of the mucin MUC2. MiR-21 also directly targets MARCKS [127] and was described to regulate mucins in the airways [128], potentially interfering with mucus secretion in the intestine. Through miRWalk, MUC2 is a predicted target of miR-16, miR-21, miR-106, miR-122, miR-146, miR-151, miR-155, miR-199 and miR-362 [119].

Goblet cells on the surface constantly secrete mucus by regulated vesicle secretion. This creates a directed flow of mucus into the lumen, supporting the clearance of the epithelial surface from debris and antigens, and plays a crucial role in maintaining the mucus layer. In contrast, the sentinel goblet cells in the higher crypts can release a massive burst of mucus through exocytosis. Upon sensing the presence of microbes, sentinel goblet cells respond through rapid exocytosis of mucins, trapping intruders to relocate before they reach the more sensitive crypts. TLRs and NLRP6-mediated inflammasomes are highly expressed in the intestine and play a crucial role in MUC2 secretion [129,130]. Indeed, NLRP-deficient mice showed a decrease in defence against pathogenic bacteria, not caused by an ineffective immune response, but rather by impaired mucus exocytosis and subsequent loss of the protective mucus layer [131].

Recent studies showed that MUC2 exocytosis in colonic goblet cells is dependent on the SNARE vesicle-associated membrane protein 8 (VAMP8) [129,132]. VAMP8 is present on mucin granules in the intestine and is important in vesicle–membrane fusion for the release of mucins [132]. VAMP8-deficient mice showed an altered mucus layer and an increased susceptibility towards DSS-induced colitis, highlighting the importance of proper exocytosis to maintain the protective mucus layer in the colon [129]. The increased susceptibility of VAMP8-deficient animals against *Entamoeba hisolytica* induced a strong pro-inflammatory response [132]. VAMP8 was also shown to play a critical role for both basal mucus secretion and exocytosis in airway goblet cells [133]. Strikingly, several miRNAs found to be elevated in IBD patients were predicted to target VAMP8. Using miRWalk, VAMP8 may be negatively affected by the upregulation of miR-21, miR-106, miR-146, miR-151, miR-155, miR-199 and miR-362 in IBD patients and thereby weaken the mucus barrier [119].

A reduction in the granule wall components BCAP31 and RAB10 was observed in UC patients [114], which were also predicted targets of the IBD-associated miR-21, miR-106, miR-146, miR-151, miR-155 (RAB10 only), miR-199 and miR-362 [119]. Interestingly, BCAP31 was shown to be directly targeted by miR-362-3p in cervical cancer [134]. Whether miR-362-3p also targets BCAP31 in goblet cells remains to be verified.

Yet another small Ras-like GTPase, RAB3A, has been identified in mucin granules of an intestinal goblet cell model [135]. RAB3A is important to regulate exocytosis [135] and is directly targeted by miR-142a-3p, promoting viral proliferation in piglets [135]. Even though miR-142 has been reported to be altered during glucocorticoid treatment for paediatric IBD [136], the direct influence on RAB3A in goblet cells remains to be determined.

The rich glycosylation of mucins in the context of diverse diseases has been recently reviewed, highlighting the altered glycosylation profiles in IBD patients [137,138,139]. The glycosylation of mucins provides protection from fast bacterial degradation and is important to maintain the gut barrier. Reduced mucus glycosylation may allow bacteria to easily penetrate the mucus layer due to removing the diffusion barrier and impairing the gradient of antimicrobial agents secreted by Paneth cells. Indeed, miR-124-3p was reported to target T-synthase, also known as C1GALT1, which catalyses the core-1 O-glycosylation of mucins. The miR-124-3p-mediated downregulation of T-synthase interferes with mucin O-glycosylation, leading to a predisposition for senile colitis [140]. The expression of C1GALT1 is dependent on a specific molecular chaperon, C1GALT1C1 (Cosmc). The dysregulation of both C1GALT1 and its chaperon Cosmc has been associated with IBD [141]. Mice deficient for C1GALT1 were reported to develop spontaneous colitis in the distal colon due to a compromised mucus layer and an increase in the exposure of commensal microbiota to the epithelium [142]. Furthermore, genome wide-association studies have linked *Cosmc* mutations with IBD. Recently, the IBD-associated miR-196b was reported to target Cosmc in patients suffering from immunoglobulin A nephropathy [143]. Through miRWalk, C1GALT1 and Cosmc might be further targeted by miRNAs as predicted binding sites were found for miR-16 (only Cosmc), miR-21, miR-106, miR-122, miR-146, miR-151, miR-155 (only C1GALT1), miR-199 and miR-362 [119].

Further insight into the role of miRNAs in the regulation of mucins can be gleaned from chronic and allergic inflammatory disorders of the airways, which are often associated with altered mucus secretion. In contrast to the colon, the protective mucus layer is mainly built from MUC5AC. MUC5AC is known to be downregulated in airways by the direct or indirect action of miR-16 [144], miR-141 [145], miR-143 [146], miR-145 [147] and miR-375 [148]. Furthermore, reduced miR-125b could be linked to increased goblet cell differentiation via SPDEF and lead to pathological hypersecretion of mucus in patients suffering from asthma [149]. The IBD-associated miR-155 was also reported to induce pathological mucus hypersecretion and modulated TH2-cell response due to its overexpression in the airways [150,151], supporting the relevance of miRNAs in mucus regulation.

Even though these predicted findings still need experimental confirmation, there is evidence that the cocktail of upregulated miRNAs in IBD patients might negatively impact the protective mucus layer by interfering with mucus function on several levels, increasing gut permeability. An overview of miRNA influence on goblet cell differentiation factors, mucus components and wall proteins of mucus granules can be seen in Figure 1.

### 3.2. Cell–Cell Interactions within the Gut Epithelium

#### 3.2.1. Intercellular Junctions

IECs exhibit remarkable cell polarisation and cell–cell adhesion, which are essential for proper barrier function [152]. Along the epithelial barrier are connection points between various IECs called intercellular junctions, which dictate paracellular transport, cell adhesion and cell–cell communication. The four types of intercellular junctions are tight junctions (TJs), adherens junctions (AJs), desmosomes/hemidesmosomes and gap junctions (GJs), all composed of distinct protein complexes used for specific roles. Of the four, TJs, AJs and desmosomes/hemidesmosomes comprise the apical junctional complex (APC) that associates with the dense network of actin and myosin filaments [152]. Most of the literature on gut permeability in IBD focuses on TJs only or has conducted predictive scans for potential intercellular junction targets of interest [153]. Here, we will review all intercellular junctions for their demonstrated importance in IEC polarity, adhesion and crosstalk, along with the regulation of junctional activity via miRNAs.

#### 3.2.2. Tight Junctions

Regarded as essential components of the gut epithelial barrier, TJs are the selective gates that control the paracellular diffusion of ions and the passage of small soluble and macromolecular compounds and restrict the entry of microbes [152,154,155]. TJ proteins are often divided into three main categories: (1) transmembrane proteins, (2) cytosolic scaffolding proteins and (3) regulatory proteins. Among the transmembrane proteins, tetraspannin-like claudins (~26 subtypes) are important pore-forming proteins that allow for selective ion permeability [155]. Cytosolic scaffolding proteins are largely attributed to zonula occludin (ZO) family proteins (ZO1, ZO2, ZO3) that interact with claudins, junction adhesion molecules (JAMs) and TJ-associated MARVEL family proteins such as occludin and tricelluin, which all govern interactions with the actin cytoskeleton [152]. These complexes work together to serve in both the “pore” and “leak” pathways dependent on the claudins that comprise TJs, for either high-capacity size/charge selection or low-capacity limited-selectivity transport, respectively [156].

In a disease state such as increased inflammation and damage to the gut epithelial barrier, TJ abundance, structure and composition are modulated, affecting the passage of microbiota and large proteins from the lumen [152]. Immunostaining for occludin and ZO1 was severely reduced in IBD tissues, particularly in IECs at the luminal surface [157]. Upregulation and redistribution of occludin, claudin-2, claudin-5 and claudin-8 were associated specifically with CD [158]. There was also increased expression of MLCK, claudin-1 and claudin-2 within patients that had active IBD [159,160]. Understanding the differences in the composition and function of TJs in healthy and diseased states could help discover underlying issues with gut permeability.

Many TJ proteins are regulated by miR-21. It has been suggested that TNF-induced miR-21 increases intestinal permeability by decreasing levels of PTEN, an inhibitor of the phosphatidylinositol 3-kinase (PI3K)–Akt signalling pathway [161]. Interestingly, intestinal symbiotic flora can upregulate miR-21-5p to target inhibitors of the PI3K–Akt, JNK–activator protein 1 (AP-1) and ERK pathways, which upregulate ADP ribosylation factor 4 (ARF4) and reduce claudin-4 and occludin, ultimately increasing barrier permeability [162]. Furthermore, miR-21 was shown to induce the degradation of Ras homolog family member B (RhoB) mRNA, leading to the depletion of occludin [48]. RhoB, part of TJ protein complexes and a target of miR-21, impacts various cellular pathways including cell cycle, apoptosis, actin organisation, cell migration and adhesion [163]. Conversely, another study suggested that miR-21 inhibition of the Rho–ROCK pathway can protect against inflammation by upregulating TJ proteins [164]. ZO1 was also shown to be regulated by miR-21 [165]. It is crucial that further research on the controversial nature of miR-21’s influence in inflammation is conducted to elaborate on its full impact.

Direct and indirect targeting of occludin and several claudins were suggested with other miRNAs. Using colon tissues from both UC patients and colitis mouse models, IL-1β-induced upregulation of miR-200c-3p decreased levels of occludin, which negatively impacted barrier function [166]. Pro-inflammatory marker IL-8 and regulator of barrier function CDH11 were shown to be directly regulated by miR-200c-3p in inflamed mucosa biopsies obtained from UC patients [52]. Additionally, TNF-induced miR-122 increased gut permeability in vitro for Caco-2 monolayers and in vivo for recycled perfused mouse intestine [167]. Moreover, miR-34 in combination with long non-coding RNA PlncRNA1 cooperatively regulated the expression of occludin and ZO1 in Caco-2 monolayers undergoing DSS-induced colitis [168]. Regarding relevant claudins, claudin-2 was recently demonstrated to be a target of miR-182-5p, whose inhibition led to increased claudin-2 and TGF-β1 expression, as well as anti-inflammatory and anti-oxidative genes [169]. The suppression of occludin and claudin-1 has been attributed to miR-874, while also targeted by miR-29b, which was shown to be sequestered via the addition of long non-coding RNA uc.173 as well as circular RNA CircHIPK3 [170,171,172], rescuing barrier function. Likewise, downregulation of claudin-8 was observed by miR-233 [173]. Interestingly, human mast cell (HMC-1)-derived exosomes enriched in miR-223 inhibited claudin-8 expression in various intestinal epithelial cell lines, destroying barrier function [174]. TJ regulation by miRNAs is abundant and demonstrates the importance of further studying their scope in IBD.

Other components of TJs, such as cytosolic scaffolding/adaptor proteins, can also be targeted by miRNAs. In T84 monolayers, ZO2 levels were impacted when inhibitors for miR-203, miR-483-3p and miR-595 were used [175]. Overexpression of miR-24 led to decreased levels of cingulin, which negatively correlates with disease severity in UC patients [176]. Additionally, TNF-induced miR-191a expression led to decreased levels of ZO1 in IEC-6 cells [177]. The suppression of aryl hydrocarbon receptor (AHR) protein and TJ proteins by miR-124 was associated with intestinal barrier disruption in both the 2,4,6-trinitrobenzene sulfonic acid (TNBS)-induced colitis mouse model and CD patient mucosal biopsy samples [178]. Interestingly, the injection of a miR-7a-5p antagomir within mice that had undergone TNBS-induced colitis increased ZO1 expression and promoted barrier recovery, potentially through downregulation of the JNK pathway [179]. MiRNA regulation of core proteins and secondary scaffolds of the TJ is frequent and thus further complicates the overall picture of the inflamed gut.

There were miRNAs that demonstrated positive roles in IBD. The expression of miRNAs that showed amelioration of disease symptoms involved miR-200b, miR-320a, miR-93 and miR-1. Through direct targeting of MLCK and c-Jun protein of the AP-1 early response transcription factor that inhibit epithelial cell proliferation, miR-200b suppressed IL-8 secretion and attenuated TJ dysfunction in vitro by preventing morphological redistribution of claudin-1 and ZO1 [180]. Transfection of T84 cells with miR-320a showed increased transepithelial resistance (TER) and JAM-A expression [181]. MiR-93 targeted protein tyrosine kinase 6 and attenuated TNF/IFNγ-induced barrier dysfunction during IBD [182]. Lastly, health benefits of salvianolic acid B, a traditional Chinese herb, were suggested to restore barrier function and TJ protein expression via the induction of miR-1 and downregulation of MLCK [183]. A further comprehensive overview of TJ regulation by miRNAs can be found in a recent review by Al-Sadi et al. (2020) [184]. Though much research has already uncovered a lot of relevant miRNAs and their targets, the non-intuitive aspect of their direct and indirect effects through downstream targeting makes it difficult to ascertain their positive or negative roles in gut permeability. An overview of miRNA influence on tight junction regulation can be seen in Figure 2 and further summarized in Table 2.

#### 3.2.3. Adherens Junctions

The major functions of AJs at the gut epithelial barrier include the stabilisation of cell–cell adhesion, the regulation of the structural actin cytoskeleton and interaction between transmembrane glycoproteins and cytosolic junctional proteins, along with linkage to intracellular signalling for transcriptional regulation [185]. The main component of the AJ along this barrier is E-cadherin, which interacts with other AJ proteins such as p120-catenin and β-catenin to link with α-catenin and regulate regional strength via peri-junctional actin assembly [186]. AJ protein complexes are also essential for the proper formation of TJs regulating the transport of molecules between IECs, which emphasises the importance of seamless regulation between intercellular junctions [152].

E-cadherin has a variety of functions in both healthy and diseased states. The dysregulation of E-cadherin leads to improper barrier function and the manifestation of IBD [187]. Total KO of E-cadherin was lethal in mice, and conditional KO at the intestinal epithelium within mouse embryos led to death within 24 h [188,189]. Reduced immunoreactive E-cadherin and α-catenin were observed in IBD tissue samples via immunofluorescence and Western blot analyses; however, no changes in mRNA levels or localisation were noted [157]. Transgenic mice have been used to study AJ dysfunction, such as the conditional deletion of E-cadherin within the intestinal epithelium, but this is likely not a primary mechanism of IBD presentation [190]. Levels of *CDH1*, which encodes E-cadherin, were downregulated in colon samples from active IBD patients [191]. Further, polymorphisms in *CDH1* have been associated with UC [192]. β-catenin-induced Wnt signalling has been implicated in IBD and has been suggested as a key regulatory pathway for barrier homeostasis [193,194]. More research is required to further elucidate modulations in E-cadherin during IBD pathogenesis.

There is some research focusing on the regulation of AJ proteins by miRNAs. Studies on the most prominent AJ proteins with patients suffering from UC or fibrosis found a negative correlation between levels of E-cadherin and miR-21 [195,196]. Recently, one study showed that exosome-derived miR-21a-5p from abnormally polarised macrophages was taken up by IECs and decreased levels of E-cadherin, exacerbating DSS-induced colitis in mice [197]. Upregulation of circRNA_102610 led to decreased levels of miR-130a-3p, promoting TGF-β1-induced EMT in human IECs and NCM460 cells and downregulating E-cadherin expression [198]. Interestingly, research using early life inflammatory stressor rat models found sustained epithelial injury through the suppression of E-cadherin via miR-155 [199]. The introduction of TGF-β1, commonly upregulated in IBD patients, to IEC-6 cells demonstrated direct targeting by miR-200b on ZEB1, a negative regulator of AJ formation and thus increased E-cadherin expression [200]. In addition to E-cadherin, other AJ proteins such as epithelial membrane protein 1 were shown to be targeted by miR-145, known to have another target cathepsin B that accumulates in IBD patients [59,201]. Still, despite the importance of AJs in gut epithelial permeability, it was to our surprise that little research has identified more interactions between key AJ regulators and their modulation via miRNAs outside of oncology, and this will be a fruitful avenue in exploring gut permeability specifically during IBD. An overview of miRNA influence on adherens junction regulation can be seen in Figure 2 and further summarized in Table 2.

**Table 2 cells-10-03358-t002:** Summary of literature demonstrating microRNA impact on intercellular junction proteins.

Intercellular Junction	Target	MicroRNA	Description	References
Tight Junction	Claudin-1	miR-29	Uc.173 and CircHIPK3 sequestered miR-29, which targets claudin-1	[171]
		miR-200b	MiR-200b supressed IL-8 secretion and thereby attenuated TJ dysfunction via claudin-1	[180]
		miR-874	Suppression of claudin-1 has been attributed to miR-874	[170]
	Claudin-2	miR-182-5p	Claudin-2 was shown to be a target of miR-182-5p	[169]
	Claudin-4	miR-21-5p	Intestinal symbiotic flora upregulated miR-21-5p to target inhibitors of the PI3K–Akt, AP-1 and ERK pathways to upregulate ARF4 and reduce claudin-4	[162]
	Claudin-8	miR-233	MiR-233 downregulated claudin-8; HMC-1 derived exosomes enriched in miR-223 inhibited claudin-8 in various intestinal epithelial cell lines	[173,174]
	Occludin	miR-21-5p	Intestinal symbiotic flora upregulated miR-21-5p to target inhibitors of the PI3K–Akt, AP-1 and ERK pathways to upregulate ARF4 and reduce occludin; also negatively affects occludin by targeting RhoB	[48,162]
		miR-29b	Uc.173 and CircHIPK3 sequestered miR-29, which targets occludin	[171]
		miR-34	PlncRNA1 and miR-34 cooperatively regulated expression of occludin in vitro during DSS-induced colitis	[168]
		miR-200c-3p	IL-1β-induced upregulation of miR-200c-3p in UC patients decreased levels of occludin	[166]
		miR-874	Suppression of occludin has been attributed to miR-874	[170]
	ZO1	miR-7a-5p	MiR-7a-5p antagomir increased ZO1 expression and promoted barrier recovery within TNBS-induced colitis models	[179]
		miR-21-5p	ZO1 shown to be indirectly regulated by miR-21	[165]
		miR-34	PlncRNA1 and miR-34 cooperatively regulated expression of ZO1 in vitro during DSS-induced colitis	[168]
		miR-191a	TNF-induced miR-191a expression led to decreased levels of ZO1 in IEC-6 cells	[177]
		miR-200b	MiR-200b supressed IL-8 secretion and thereby attenuated TJ dysfunction via ZO1	[180]
	ZO2	miR-203	ZO2 levels were impacted when inhibitors for miR-203, miR-483-3p and miR-595 were used on T84 monolayers	[175]
		miR-483		
		miR-595		
	Cingulin	miR-24	Overexpression of miR-24 led to decreased levels of cingulin, which negatively correlated with disease severity in UC patients	[176]
	JAM-A	miR-320a	Elevated miR-320a increased TER and JAM-A expression in T84 cells	[181]
	Other	miR-1	Health benefits of salvianolic acid B were suggested to restore barrier function and TJ protein expression via induction of miR-1	[183]
		miR-93	MiR-93 targeted protein tyrosine kinase 6 and attenuated TNF/IFNγ-induced barrier dysfunction during IBD	[182]
		miR-122	TNF-induced miR-122 increased gut permeability	[167]
		miR-124	Suppression of AHR protein and TJ proteins by miR-124 was associated with intestinal barrier disruption	[178]
Adherens Junctions	E-Cadherin	miR-21	A negative correlation between levels of E-cadherin and miR-21 was found in UC patients; exosome-derived miR-21a-5p from abnormally polarised macrophages decreased levels of E-cadherin in the epithelium, exacerbating DSS-induced colitis in mice	[195,196,197]
		miR-130a-3p	Upregulated circRNA_102610 decreased miR-130a-3p, promoted TGF-β1-induced EMT in vitro and downregulated E-cadherin expression	[198]
		miR-155	E-cadherin was negatively affected by miR-155, leading to epithelial injury in early-life inflammatory stressor rat models	[199]
		miR-200b	MiR-200b directly targets ZEB1, a negative regulator of AJ formation, and thus increased E-cadherin expression	[200]
	Epithelial Membrane Protein 1	miR-145	Other AJ proteins such as epithelial membrane protein 1 were shown to be targeted by miR-145	[59]

#### 3.2.4. Desmosomes and Hemidesmosomes

The primary role of desmosomes and hemidesmosomes at the gut epithelial barrier is to resist shearing forces, acting as anchoring points to the basal membrane, and to establish a continuum layer of cells through linkage via membrane proteins (e.g., desmogleins, desmocollins and desmoplakins) to armadillo repeat family proteins (plakoglobin and plakophilin) with desmoplakin and finally intermediate filaments [202]. Interestingly, desmocollins and desmogleins are part of the larger cadherin family, and thus have similar characteristics to AJs; however, they possess the unique ability of calcium-independent hyper-adhesiveness [203]. Since desmosome expression is tissue-specific, only the membrane proteins desmoglein-2 and desmocollin-2 are expressed in the gut [202]. For hemidesmosomes, proteins such as BP230 and plectin are the most well-known, and are implicated in the organisation of cytoskeletal elements [204], as well as integrin α6β4 [205].

Although part of the APC, desmosomes/hemidesmosomes have been largely overshadowed by TJs and AJs, particularly in their relevance to IBD [202]. Total KO of desmoglein-2 led to decreased levels of claudin-1 and occludin, increasing intestinal permeability [206]. Epithelial-specific KO of desmocollin-2 showed no increased barrier permeability during DSS-induced colitis; however, another study showed that a conditional-inducible KO of desmocollin-2 had impaired mucosal repair after recovery from DSS-induced colitis [207]. Additionally, desmoglein-2 regulated claudin-2 expression via the sequestration of PI3-K in IECs of mice during DSS-induced colitis [208]. Desmosomal staining of epithelial cells within patients afflicted with either CD or UC was reduced in a manner dependent on the severity of inflammation, complemented by decreased protein abundances of desmoglein-2 and desmocollin-2 via Western blot [157]. Other studies of CD patient cohorts of various levels of disease severity confirmed the decrease and abnormal distribution of desmoglein-2 and desmocollin-2 within inflamed tissue [209]. For hemidesmosomes, total and conditional KO mice for integrin α6β4 were generated and led to spontaneous colitis caused by detachment from the basal membrane and the underlying lamina propria that induces an IL-1β/IL-18 pro-inflammatory response [210]. Despite their clear roles in maintaining barrier integrity, no studies on the regulation of desmosomes/hemidesmosomes by miRNAs could be found, and therefore more research should be conducted to provide knowledge on their function in IBD.

#### 3.2.5. Gap Junctions

Less structural but rather communicative components of the gut epithelial barrier are GJs. Prevalent between adjacent IECs are GJs made of intracellular plasma membrane channels allowing for cell–cell communication via the passage of biologically important ions and small metabolites [211]. GJs are typically composed of homologous proteins called connexins (~21 in humans) that can form up to six homomeric/homotypic or heteromeric/heterotypic connexons, differing in content and spatial arrangement depending on their permeability roles when binding to adjacent cell connexons to form intercellular channels [212]. Therefore, a core function of these junctions is to share metabolic demands across groups of cells and buffer gradients in space, nutrients and signalling molecules.

A major role of GJs at the gut epithelial barrier is for effective crosstalk among different cell types found at this region. Mouse macrophages were shown to communicate with IECs via the use of GJs [213,214]. Additionally, TLR4-mediated crosstalk between macrophages and IECs results in IL-10 production, which could regulate barrier integrity, likely requiring the coordinated functioning of GJs [215]. Co-cultures with IECs and THP-1 macrophages demonstrated the importance of these heteromeric communication channels between different cell types [216]. In Caco-2 cell monolayers, stable overexpression of connexin-26 increased claudin-4 expression and TER measurements during monolayer disruption with oleic acid and taurocholic acid [217]. Regarding IBD, immunohistochemistry analysis of common connexins such as connexin-26 and connexin-43 demonstrated lower expression at the apical end of IECs with greater localisation at the basolateral end in IBD tissue, suggesting a role in communication with infiltrating macrophages during a disease state [216]. Co-localisation of the prominent connexin-43 with other intercellular junction proteins important at the epithelial barrier such as E-cadherin and ZO1 is lost in IBD tissues [216]. TLR2-mediated mucosal healing after acute intestinal barrier damage modulated levels of connexin-43 [218]. Interestingly, miRNAs have been shown to pass through gap junctions of adjacent cells in a connexin-dependent manner that favoured connexons primarily made of connexin-43, further demonstrating a synchronised coordination of miRNA regulation at the gut epithelial barrier [219]. Therefore, furthering work on the coordination of TLRs and GJ proteins in IEC communication is necessary to determine the overall response to microbiota in both healthy and diseased states.

## 4. Conclusions

Permeability is an important feature at the gut epithelial barrier, since many essential nutrients are absorbed at this interface, along with normal microbiota-priming of the gut-associated immune system via the passage of antigens. However, too much permeability results in chronic inflammation, leading to debilitating disease. Under these circumstances, cellular regulatory processes are modulated, particularly through the expression of miRNAs. Previous reviews on miRNA impact in gut disease typically focused on clinical association studies and gut immunity. The current review focused on miRNA-associated physical cellular factors contributing to IBD, specifically the initial protective mucus layer and the interactions between IECs within the underlying gut epithelium. Many studies have discovered associations with differential expression of miRNAs and components within these barriers, but few have determined mechanistically the direct and indirect targeting of these associations. There are multiple shared miRNAs that govern the specific regulation of protective mucins, along with the interactions among IECs via the four intercellular junctions. Clearly, further research is required in order to establish holistic miRNA regulation of these features, which could lead to further development of biomarkers and therapeutics preventing impaired permeability during IBD.

## Figures and Tables

**Figure 1 cells-10-03358-f001:**
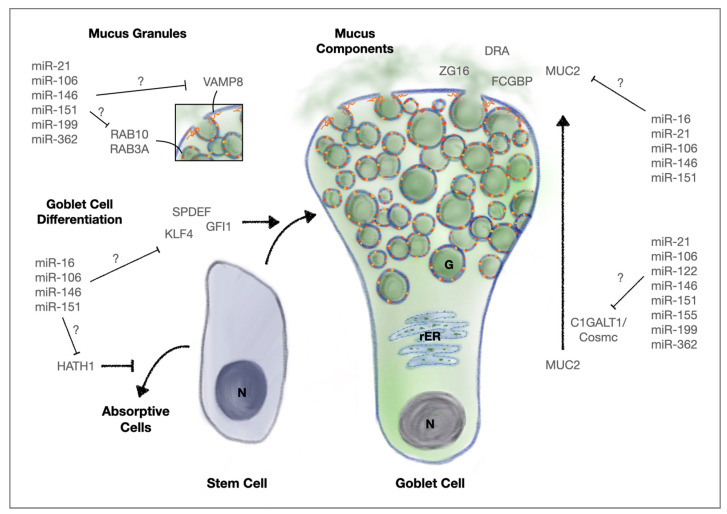
Predicted influence on the mucus layer by IBD-associated miRNAs through targeting goblet cell differentiation factors, mucus components and wall proteins of mucus granules. N: nucleus, rER: rough endoplasmic reticulum, G: mucin granules.

**Figure 2 cells-10-03358-f002:**
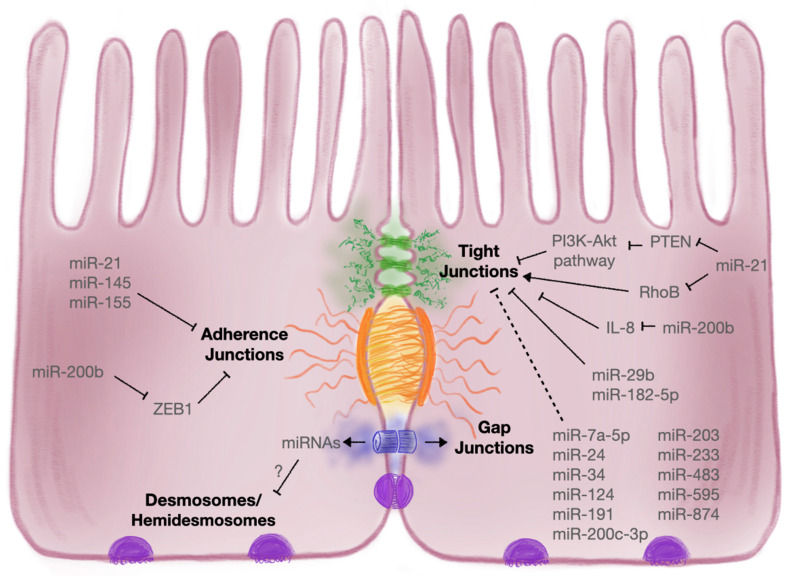
Overview of altered miRNAs during IBD shown to target intercellular junction proteins and thereby weakening the gut barrier. Dashed line: indirect target, solid line: direct target.

**Table 1 cells-10-03358-t001:** Summary of altered microRNA expression patterns in IBD.

MicroRNA	Expression Level	Sample	Biomarker	References
let-7f	upregulated	colonic tissue	diagnosed UC patients	[48]
miR-16	downregulated	colonic tissue; plasma	active UC; diagnosis of CD	[48,49]
	upregulated	serum and colonic mucosa; blood; biopsy; colonic tissue	canine IBD model; diagnosed IBD patients; diagnosed UC patients	[16,38,40,48]
miR-20b	differential pattern	colonic mucosa	active vs. quiescence UC	[47]
miR-21	upregulated	colonic tissue; blood; serum; saliva	diagnosed UC patients; diagnosed IBD patients; canine IBD model	[16,36,38,40,48]
miR-23a	upregulated	colonic tissue	diagnosed UC patients	[48]
miR-24	upregulated	colonic tissue	diagnosed UC patients	[48]
miR-26b	differential pattern	colonic mucosa	active vs. quiescence UC	[47]
miR-29a	upregulated	colonic tissue	diagnosed UC patients	[48]
miR-31	upregulated	colonic mucosa, saliva	diagnosed IBD and UC patients	[36,37]
miR-31-5p	differential pattern	colonic tissue	diagnostic marker for CD	[51]
miR-98	differential pattern	colonic mucosa	active vs. quiescence UC	[47]
miR-99a	differential pattern	colonic mucosa	active vs. quiescence UC	[47]
miR-101	upregulated	saliva	CD	[36]
miR-106a	upregulated	blood/biopsy	diagnosed IBD patients	[16,40,41]
miR-122	upregulated	blood/biopsy; serum and colonic mucosa	diagnosed IBD patients; canine IBD model	[16,38,40]
miR-126	upregulated	colonic tissue	diagnosed UC patients	[48]
miR-142-3p	upregulated	saliva	UC	[36]
miR-142-5p	differential pattern	serum	active vs. quiescence CD	[33]
	downregulated	saliva	UC	[36]
miR-146a	upregulated	colonic mucosa; serum	diagnosed IBD patients; canine IBD model	[37,38]
miR-147	upregulated	serum and colonic mucosa	canine IBD model	[38]
miR-150	differential pattern	colonic tissue	non-inflamed UC vs. non-inflamed CD	[46]
miR-151-5p	upregulated	blood/biopsy	diagnosed IBD patients	[16,39,40]
miR-155	upregulated	blood/biopsy	diagnosed IBD patients	[16,39,40]
miR-192	upregulated	serum; colonic tissue	canine IBD model; active UC	[38,48,49]
miR-192	downregulated	colonic tissue	diagnosed UC patients	[48]
miR-195	upregulated	colonic tissue	diagnosed UC patients	[48]
miR-196b	differential pattern	colonic tissue	non-inflamed UC vs. non-inflamed CD	[46]
miR-199a-3p	differential pattern	colonic tissue	non-inflamed UC vs. non-inflamed CD	[46]
miR-199a-5p	upregulated	blood/biopsy	diagnosed IBD patients	[16,40]
miR-199b-5p	differential pattern	colonic tissue	non-inflamed UC vs. non-inflamed CD	[46]
miR-203	differential pattern	colonic tissue	active vs. quiescence UC; diagnostic marker for CD	[47,51]
miR-206	upregulated	colonic mucosa	diagnosed IBD patients	[37]
miR-223	upregulated	serum	canine IBD model	[38]
	differential pattern	colonic tissue	non-inflamed UC vs. non-inflamed CD	[46]
miR-320	upregulated	blood/biopsy	diagnosed IBD patients	[16,39,40]
miR-320a	differential pattern	colonic tissue	non-inflamed UC vs. non-inflamed CD	[46]
miR-362-3p	upregulated	blood/biopsy	diagnosed IBD patients	[16,40,41]
miR-375	downregulated	colonic tissue	diagnosed UC patients	[48]
miR-422b	downregulated	colonic tissue	diagnosed UC patients	[48]
miR-424	upregulated	colonic mucosa	diagnosed IBD patients	[37]
miR-595	differential pattern	serum	active vs. quiescence CD	[33]
	upregulated	serum	non-specific biomarker for IBD	[33]
miR-1246	upregulated	serum	non-specific biomarker for IBD	[33]
	differential pattern	serum	active vs. quiescence CD and UC	[33]
miR-1307-3p	upregulated	blood (CD4+ T-cells)	disease progression in IBD	[35]
miR-3615	upregulated	blood (CD4+ T-cells)	disease progression in IBD	[35]
miR-4284	downregulated	colonic tissue	active UC	[50]
miR-4792	expression	blood (CD4+ T-cells)	disease progression in IBD	[35]

## Data Availability

Not Applicable.

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
