# Peer review of "The Impact of MicroRNAs during Inflammatory Bowel Disease: Effects on the Mucus Layer and Intercellular Junctions for Gut Permeability"

_cells, 2021, doi:10.3390/cells10123358_

Round 1
Reviewer 1 Report
In the present review, Stiegeler et al., provided an overview of the available knowledge on the impact of miRNAs on goblet cell secretion and mucin structure occurring in the inflammatory bowel diseases (IBDs).
The manuscript is interesting. However, there are some points of concern, which should be properly addressed to further improve the quality of the manuscript.
MINOR POINTS:
_ The authors need to carefully review the English style and the expressions. The authors are recommended to have a native english speaker spellcheck the manuscript thoroughly. Several concepts are not clearly discussed.
_ What's the difference between your review and other reviews in the field of mi-RNAs in IBDs?
_ In some points (i.e., page 5 lines 204- 215), the review seems to be a simple list of altered mi-RNAs. The authors should better explain the pathophysiological significance of these alterations in the setting of IBD.
_ Please replace the wording “TNF-α” with “TNF” throughout the manuscript
_ Please insert “of” page 3 line 105 in “diagnosing of the two”
_ The authors should insert the table number (i.e. Table 1) and figure number (Figure 1, Figure 2..) at the end of sentence to help the reader
_ A table that summarize the alterations of mi-RNA expression in IBD discussed on page 3 lines 113-129 should be inserted to help the readers
_ The authors should add the full name of the abbreviation “IECs” on page 4 line 166; while delete the full name “intestinal epithelial cells” on page 4 line 184
Author Response
MINOR POINTS:
1. "The authors need to carefully review the English style and the expressions. The authors are recommended to have a native english speaker spellcheck the manuscript thoroughly. Several concepts are not clearly discussed."
Response: This has now been resolved.
2. "What's the difference between your review and other reviews in the field of mi-RNAs in IBDs?"
Response: Previous reviews on microRNAs and pathogenesis generally stem from oncology or autoimmunity. There are some reviews which have narrowed focus on microRNA impact in IBD, yet still with a major focus on general association studies and gut immunity. Our review does include those important factors, but more specifically summarizes the contribution of microRNAs to the weakening/dysfunction of the gut from the perspective of what could be considered the initial physical barrier; the mucus layer and the intestinal epithelial cells (via functioning of intercellular junctions). Our review includes lists of modulated miRNAs in these contexts, and therefore a statement on the novelty of this review has been included in the Conclusion section.
3."In some points (i.e., page 5 lines 204- 215), the review seems to be a simple list of altered mi-RNAs. The authors should better explain the pathophysiological significance of these alterations in the setting of IBD."
Response: Resolved. The Authors are somewhat unclear what is being requested as the pathophysiological significance of mentioned microRNA alterations have been explained. Some lists are of predicted microRNAs or microRNAs whose levels have been modulated and found to affect gene expression of mucin/junction components specifically.
4. "Please replace the wording “TNF-α” with “TNF” throughout the manuscript"
Response: Resolved.
5. "Please insert “of” page 3 line 105 in “diagnosing of the two”
Response: Resolved.
6. "The authors should insert the table number (i.e. Table 1) and figure number (Figure 1, Figure 2..) at the end of sentence to help the reader"
Response: Resolved.
7. "A table that summarize the alterations of mi-RNA expression in IBD discussed on page 3 lines 113-129 should be inserted to help the readers"
Response: Resolved.
8. "The authors should add the full name of the abbreviation “IECs” on page 4 line 166; while delete the full name “intestinal epithelial cells” on page 4 line 184"
Response: Resolved.
Reviewer 2 Report
The review by Stiegeler et al, analyzes the impact of MicrRNA on goblet cell secretion and mucin structure, along with the proper function of various junctional proteins involved in paracellular transport, cell adhesion and communication on the inflammatory bowel disease.
The authors describe how a dis-regulated miRNA expression can contribute to the development of gut inflammation in order to propose therapeutic strategies to ameliorate gut permeability.
The introduction shows the main characteristics of miRNA, their structures and targets, and how are involved in gastrointestinal health.
In the second part miRNA are related to the various intestinal diseases as Chron’s disease (CD) and ulcerative colitis (UC) and how they impact on gut immunity, permeability, and mucus secretion.
The figures provided are clear, helping to recapitulate the miRNA network in gut structure and function.
The table gives an insight into the major studies demonstrating microRNA's impact on intercellular junction proteins.
The review is well structured and well written and references are adequate.
Author Response
Response: The Authors thank the reviewer for these positive comments and feedback.